# Sexual violence victimisation and response among university students in sub-Saharan Africa: a scoping review protocol

Ester Steven Mzilangwe [1,2] Rehema Chande Mallya [3] Marie Lindkvist [1] Sylvia Kaaya [2] Faustine Kyungu Nkulu Kalengayi [1]

¹Department of Epidemiology and Global Health, Umea University, Umea, Sweden
²Department of Psychiatry and Mental Health, Muhimbili University of Health and Allied Sciences, Dar es Salaam, Tanzania, United Republic of
³Directorate of Library Services, Muhimbili University of Health and Allied Sciences, Dar es Salaam, Tanzania, United Republic of

**Correspondence to**
Dr Ester Steven Mzilangwe;
ester.mzilangwe@umu.se

## ABSTRACT

**Introduction** Sexual violence (SV) is highly prevalent among university campuses across the globe, despite of several initiatives implemented to address it. Several studies have been published focusing on various aspects of SV on campuses. However, no review has been retrieved from the Joanna Briggs Institute (JBI) Database, Cochrane Library or Ovid examining evidence synthesis on prevalence, risk factors, victims and perpetrators, policies, laws and universities responses to SV in sub-Saharan Africa (SSA). This review aims to map the existing literature on SV victimisation among university students in SSA, related response strategies, and identify gaps in the evidence.

**Methods and analysis** This review will follow JBI guidelines and will be conducted from 1 July 2023 to 31 December 2023. A team of five reviewers will screen eligible documents and articles for relevance from various data sources including electronic databases such as MEDLINE, EMBASE, PsycINFO, CINAHL, Google Scholar, PubMed and websites for government and agencies. Standard information for each study will be collected and a common analytical framework for all the primary documents will be conducted.

**Ethics and dissemination** This review will involve analysis of published data only and therefore does not require ethics approval. The results will be published in a peer-reviewed journal.

**Registration** This review has been registered with the Open Science Framework.

## STRENGTHS AND LIMITATIONS OF THIS STUDY

⇒ The review approach will cover information from broad sources, therefore, providing broader synthetic information on sexual violence victimisation and response mechanisms on university campuses in sub-Saharan Africa.
⇒ Studies and documents that will be included in this review are conducted and prepared by other people and therefore may include the risk of bias.
⇒ The review will be limited to articles and documents published in the English and French languages only and therefore will lack evidence from documents and articles published in other languages.

## INTRODUCTION

Sexual violence (SV) is a serious global public health problem and a violation of sexual rights affecting millions worldwide.[1] SV is defined as any sexual act, attempt to obtain a sexual act, unwanted sexual comments or advances or acts to traffic, or otherwise directed, against a person's sexuality using coercion, by any person regardless of their relationship to the victim, in any setting, including but not limited to home and work.[2] SV is commonly reported on university campuses around the world with alarmingly high estimates.[3]

Globally, the prevalence rate of sexual assault on undergraduates has been estimated at 10.3% in women and 3.1% in men.[4] Studies from sub-Saharan Africa (SSA) indicate that the prevalence of various forms of SV and sexual harassment experienced by university students in five countries vary within and across countries from 14.3% to 78.2%.[5–9]

There are several risk factors for SV victimisation in university settings. Young women may have sex with a teacher or someone else they are dependent on because they need to pass a course to get financial support, in that way, promotes their subordination and dependency on men.[10 11] Individual factors like young age, alcohol and drug consumption, year of study and living out of campus have also been associated with SV victimisation.[12] Other factors include previous SV victimisation, multiple sex partners and losing a parent before age 2.[5 8 9 12 13] Weak monitoring systems, lack of guidelines and unprofessional administration, and community attitudes are some of the institutional and community-level factors that are associated with SV victimisation.[12]

Several initiatives are implemented to address SV in universities; these include

BMJ Group

development of policies and guidelines to prevent and respond to SV, accessible and trauma-informed services for supporting student disclosure and advocacy.[14] However, these initiatives meet the number of challenges including the unintended consequences of mandatory reporting that may include secondary victimisation and stigma, and lack of knowledge about available resources on the campus.[3 15]

Despite the reported high rates of SV victimisation on campuses in SSA countries,[5–9] the available evidence has not been synthesised to provide an overall picture and a clear understanding of not only the extent of the problem but also knowledge of potential victims, and perpetrators and what strategies/interventions have been implemented to respond to SV in SSA universities. Moreover, at the time of writing this protocol, there was no retrieved evidence synthesis or review of existing evidence in the form of scoping or systematic review conducted in the area of SV victimisation and response among university students across SAA countries from the Joanna Briggs Institute (JBI) Database of Systematic Reviews and Implementation Reports, Cochrane Library or MEDLINE via Ovid website. In response to this gap, this scoping review will conduct a mapping of available evidence on SV victimisation on campuses and different strategies used to respond to/address it in SSA universities. In addition, this scoping review will provide preliminary information to guide our research agenda.

### Aim, objectives and research questions

The overall aim of this scoping review is to map existing literature on SV victimisation among university students in SSA and related response strategies and to identify gaps in the evidence. This review will provide an evidence basis for specific recommendations in future work and clearly state gaps in the literature needed to guide the research agenda.

The specific objectives of this scoping review are:

► Map existing literature on SV prevalence, types, risk factors and common victims and perpetrators of SV among university students on campuses in SSA,

► Describe different laws, policies, strategies and actions taken to address and prevent SV on campuses and

► Identify evidence gaps.

The main questions guiding this scoping are:

1. What does existing literature tell us about SV prevalence, types, risk factors and common victims and perpetrators of SV on campuses in SSA?

2. What are the various laws, policies, strategies and actions that have been implemented to address and prevent SV against university students on campuses in SSA?

3. What is the gap in knowledge of the existing literature on SV victimisation and response among university students on campuses in SSA?

## METHODS

The review has been registered with the Open Science Framework and will be conducted from 1 July 2023 to 31 December 2023. The review will follow JBI guidelines for scoping reviews.[16] These guidelines were developed from the framework proposed by Arksey and O'Malley's[17] and further enhanced by Levac, Colquhoun and O'Brien who provided more details of what should be done in each stage of the review process.[18] The JBI guidelines not only propose a detailed and comprehensive step-by-step process of conducting scoping review but also they are highly recommended to ensure a high quality review in a clear and transparent process.[19] The review will be reported according to the Preferred Reporting Items for Systematic Reviews, and Meta-analyses extension for scoping review checklist.[20] With regards to this, the steps towards conducting this review will include (1) search strategy, (2) study selection, (3) charting the data, and (4) collating, (5) summarising, and (6) reporting the results. The details of what will be done in each step are explained below.

### Information sources and search strategy

The reviewers will use the JBI-proposed three-step search strategy to identify relevant articles for this study.[16] The reviewers will first start by searching from two databases (MEDLINE and CINAHL) using the title of this review. The reviewers will then analyse the text words contained in the title and abstract of retrieved papers, and of the index, terms used to describe the articles. The second step will focus on all index terms and identified keywords from the titles and abstracts of the retrieved papers. These index terms will then be used to search all selected information sources including electronic databases such as MEDLINE, EMBASE, PsycINFO, CINAHL, Google Scholar and PubMed. The complete electronic search strategy for all databases can be found under online supplemental materials, further search strategy details across bibliographic databases are available on request from the first author. Additional sources, search terms and keywords may be discovered and incorporated into the search strategy. The reviewers will also search for other sources and other documents including government policies and guidelines from government websites and local and international agencies that will be identified during the search process. In the third step, reviewers will use the reference list of all selected articles to search for additional studies and include them in the list. The list of all identified studies and documents will be uploaded to the EndNote reference manager software V.20.5 (Clarivate), and duplicates will be removed. To minimise reporting bias, this scoping review will be conducted by two independent primary reviewers and two secondary reviewers. Reviewers will contact authors of primary studies, reviews and policy

documents for further information when additional information is required.

## Eligibility criteria

As proposed by the JBI guidelines,[16] we used the population, concepts and context (PCC) approach to formulate our inclusion criteria for the studies that will be reviewed.

### Inclusion criteria

Population: The target population for this review will be the human subjects who are university students, both men and women, aged 18 years and above.

Concept: Our core concept for this review is SV and we will focus on its/in relation to prevalence, types, risk factors, common victims and perpetrators as well as laws, policies, strategies and actions taken to address SV on campuses.

Context: This review will be limited to studies conducted in universities in SSA and policy documents addressing sexual violence in universities in SSA, from the inception until 31 December 2023.

### Exclusion criteria

Studies conducted in settings other than universities, that focused on areas other than SSA, populations other than students not focusing on SV in relation to the concepts listed above and those published in another language other than English or French will be excluded.

## Study selection

Reviewers will assess the retrieved articles for relevance using the inclusion criteria. This will be done in two steps, first by screening the title and abstracts and then by full text whereby copies of full articles will be retrieved. A list of those studies that will meet the inclusion criteria will be selected. At this stage, reviewers will convene and jointly discuss any discrepancies between the two lists. Secondary reviewers will be invited and consulted to resolve any discrepancies before moving to the next stage. The reviewers will then read the full article for assessment against the inclusion criteria for the review. The detailed flow chart of all studies that were included and excluded, with clear reasons for exclusion will be developed and included in our report. Any disagreements that arise between the reviewers at this stage of the selection process will be resolved through discussion with secondary reviewers.

## Charting the data

Reviewers will start by reading the primary research articles and policy documents and collecting standard information about each article. Reviewers will then extract information and enter it in the prepared data charting form using the database programme Excel. Information that will be initially entered in the data charting form will include the author(s), year of publication, study location (country), study population, aims of the study, methodology, data collection tools, outcome measures and key findings. Thereafter, the reviewers will make a broader narrative review to analyse information based on the

context of the study findings and their conclusion. The charts will contain both numerical summaries and qualitative thematic analysis findings.

## Collating, summarising and reporting the results

In this stage, reviewers will focus on identifying the implications of the study findings for policy, practice or research. All articles included in the review will be screened for results. We will develop a thematic framework that will enable us to present a narrative account of all extracted information from the articles based on overlaps and diversities of the findings to answer our research questions.

## Summarising

Depending on the findings from the studies we will select the best way to organise and summarise that data, these can include figures, tables and evidence gaps maps, in a diagrammatic or tabular form, and/or in a descriptive format that aligns with our objectives and elements of the PCC as mentioned above. Descriptive analysis of the extracted data will be done, the type of descriptive analysis will depend more on the type of extracted data.

## Reporting the results

The decision of which method of reporting to be used will be done by involving all reviewers in the team. The outcome of this review is expected to include findings on SV victimisation among university students in SSA countries and related information that will be mapped in our findings like characteristics of victims and perpetrators. Another outcome will be response systems following SV victimisation and the characteristics of the response system, the structures and users and the strength and gaps. The policy documents and guidelines will also be reviewed, and details of implementations that have been done, the strengths, as well as the gaps, will be summarised in this review.

## ETHICS AND DISSEMINATION

Since this review will only involve analysis of published data, it does not require ethics approval. We anticipate that these results will be disseminated through publication in a peer-reviewed journal and conferences targeting researchers, programmers, policymakers, responders and university students across the continent. The data set will be made available at the Umea University and the Muhimbili University of Health and Allied Science repositories. We expect that the findings of this scoping review will provide important information on SV prevalence, types, risk factors, common victims and perpetrators, laws, policies, strategies and actions taken to address SV on campuses in SSA that will guide future work. The findings will further delimit the gaps in the published data to guide the forthcoming research agenda in SSA.

**Contributors** ESM conceptualised the review approach and wrote the first draft of the manuscript. RCM reviewed and provided inputs in the methods section. FKNK

identified the framework and provided general guidance in all sections. ML and SK reviewed the framework and provided inputs on the scope of the review and methods. All authors read and approved the final manuscript.

**Funding** This work was supported by the Erling Persson's Foundation (EP) through the Department of Epidemiology and Global Health (EpiGH), Umeå University, for training and the Muhimbili University of Health and Allied Sciences by providing a protected time for the corresponding author to conduct this review.

**Competing interests** None declared.

**Patient and public involvement** Patients and/or the public were not involved in the design, or conduct, or reporting, or dissemination plans of this research.

**Patient consent for publication** Not applicable.

**Provenance and peer review** Not commissioned; externally peer reviewed.

**ORCID iDs**
Ester Steven Mzilangwe http://orcid.org/0009-0009-9890-4916
Rehema Chande Mallya http://orcid.org/0000-0001-6342-7712
Marie Lindkvist http://orcid.org/0009-0004-2568-8136
Sylvia Kaaya http://orcid.org/0000-0001-6732-3590
Faustine Kyungu Nkulu Kalengayi http://orcid.org/0000-0002-2061-323X

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
