## [Reviewer comments · BMJ Open]

ARTICLE DETAILS

TITLE (PROVISIONAL)	Sexual violence victimization and response among university students in sub-Saharan Africa: A scoping review protocol
AUTHORS	Mzilangwe, Ester; Chande Mallya, Rehema; Lindkvist, Marie; Kaaya, Sylvia; Nkulu Kalengayi, Faustine Kyungu

VERSION 1 – REVIEW

REVIEWER	Pillai, Divya Centre for Chronic Conditions and Injuries
REVIEW RETURNED	26-Sep-2023

GENERAL COMMENTS	Thank you for this opportunity. This is going to be an important study considering the burden of sexual violence victimization or exploitation among students. I appreciate the researchers for taking the initiative to map the prevalence, risk factors, laws and policies and gaps in the evidence through an extensive review. But, would like to suggest a few things that may help in improving the quality of the data and the manuscript. The introduction seem to be very lengthy and not so organized. It can be in align with the research questions or the objectives. There are statements without figures. Scientifically these can be avoided. Also, a few sections sentences seem to be very lengthy (3-4 lines) and vague. Contradicting statements (highlighted) can be avoided. In the methodology section can be more elaborated as there are clear guidelines given in the JBI manual. Eligibility criteria can be revised. Collating, analysing and summarizing the results needs a major revision. The authors hasnt detailed the statistical analysis plan for this review. Also, not given in contrast with the objectives. The reader has to go back and check the objectives to understand this section. As the researchers are already initiated the review activities, it would be great to note down the anticipated challenges and limitations of the review. Disclaimer: I havent checked the manuscript for plagiarism and references.
---

	(The reviewer provided a marked copy with additional comments. Please contact the publisher for full details.)
--	--

REVIEWER	Mazhambe, Raymond SRHR Africa Trust
REVIEW RETURNED	20-Oct-2023

GENERAL COMMENTS	The paper provides a very clear outline of carrying out the scoping review for the "Sexual violence victimization and response among university students in sub-Saharan Africa." The authors have provided an overview on the proposed methods and approaches to be adopted and clarity of thought on how to carry out the review process. It will be important for the authors add more clarity on the adoption of the Joanna Briggs Institute guidelines and provide a more justification on "why" this method and share the benefits of using this method in scoping reviews to readers who are not familiar with this method. In the introduction section, the paper provides good examples of the cases of SV in universities across the UK and USA and the different levels of power in the cases of SV among university students and acknowledges that there is few cases of SV highlighted in SSA universities and then later highlights that there is no evidence of SV cases in SSA universities with acknowledgement of journal sources. I would recommend considering revision of that section for consistency and alignment. There is also need to write JBI in full for the first time (JBI Database of Systematic Reviews) so that a reader who is not familiar with the acronym is guided accordingly. References - to also consider adding recent scholarly sources/citations. Overall, I would recommend the publication of this scoping review if the minor revisions are addressed.
---

VERSION 1 – AUTHOR RESPONSE

REVIEWERS COMMENTS	RESPONSE
Reviewer: 1	
This is going to be an important study considering the burden of sexual violence victimization or exploitation among students. I appreciate the researchers for taking the initiative to map the prevalence, risk factors, laws and policies and gaps in the evidence through an extensive review. But, would like to suggest a few things that may help in improving the quality of the data and the manuscript.	Thank you, this is appreciated.

 The introduction seem to be very lengthy and not so organized. It can be in align with the research questions or the objectives. There are statements without figures. Scientifically these can be avoided. Also, a few sections sentences seem to be very lengthy (3-4 lines) and vague. 	Thank you for this observation. We have amended the entire paragraph and summarized the important areas. Moreover, we have reduced the lengthy sentences as well.
 Contradicting statements (highlighted) can be avoided. 	We could not access the document with highlights; however, we went through the document and re-write the statement that indicate contradiction. We hope we manage to address this important observation.
 In the methodology section can be more elaborated as there are clear guidelines given in the JBI manual. 	Thank you for this observation. We added more details to elaborate the review process.
 Eligibility criteria can be revised. 	We revised eligibility criteria and included French articles since one of the authors is fluent in this language.
 Collating, analysing and summarizing the results needs a major revision. The authors hasnt detailed the statistical analysis plan for this review. Also, not given in contrast with the objectives. The reader has to go back and check the objectives to understand this section. 	We anticipate conducting descriptive analysis of the extracted data however, this will depend on the type of data that will be extracted. Since this is a scoping review, we do not anticipate doing any statistical tests apart from descriptive summaries as we have mentioned.
 As the researchers are already initiated the review activities, it would be great to note down the anticipated challenges and limitations of the review. 	Anticipated challenges and limitation

Reviewer: 2	
The paper provides a very clear outline of carrying out the scoping review for the "Sexual violence victimization and response among university students in sub-Saharan Africa." The authors have provided an overview on the proposed methods and approaches to be adopted and clarity of thought on how to carry out the review process.	Thank you
 It will be important for the authors add more clarity on the adoption of the Joanna Briggs Institute guidelines and provide a more justification on "why" this method and share the benefits of using this method in scoping reviews to readers who are not familiar with this method. 	We have noted this input and have included the statement to motivate why we selected this method.
 In the introduction section, the paper provides good examples of the cases of SV in universities across the UK and USA and the different levels of power in the cases of SV among university students and 	Thank you for this observation. We have amended the entire paragraph and summarized the

acknowledges that there is few cases of SV highlighted in SSA universities and then later highlights that there is no evidence of SV cases in SSA universities with acknowledgement of journal sources. I would recommend considering revision of that section for consistency and alignment.	important highlights. The statement on no available evidence focuses on the lack of review of the available evidence like scoping and systematic reviews, but we did not mean to indicate that there is no evidence of SV cases in SSA universities.
 There is also need to write JBI in full for the first time (JBI Database of Systematic Reviews) so that a reader who is not familiar with the acronym is guided accordingly. 	Thank you for this insight, we have corrected this in the abstract session.
 References - to also consider adding recent scholarly sources/citations. 	We have changed some articles cited to more recent sources.
Overall, I would recommend the publication of this scoping review if the minor revisions are addressed.	Thank you for this recommendation

VERSION 2 – REVIEW

REVIEWER	Pillai, Divya Centre for Chronic Conditions and Injuries
REVIEW RETURNED	12-Dec-2023

GENERAL COMMENTS	Thank you for the opportunity and accepting the suggestions given in the previous review. A few minor revisions like reframing the abstract introduction and other minor suggestions as commented will be appreciated. It is a great initiative and congratulations to the researchers as well as the BMJ for disseminating the research across the globe. (The reviewer provided a marked copy with additional comments. Please contact the publisher for full details.)
--

REVIEWER	Mazhambe, Raymond SRHR Africa Trust
REVIEW RETURNED	17-Dec-2023

GENERAL COMMENTS	The paper is reading well and has shown great changes from the recommendations provided in the first review submission. I have two sections that i have noted which require alignment:  1. Section on Specific Objectives and Research Questions must be aligned and speak to each other. Specific objectives should translate into the specific research questions. 2. Section on ETHICS AND DISSEMINATION - i have noted Tanzania appearing for the first time and please provide a justification on why the findings are important for Tanzania. If these issues are addressed, the manuscript is ready for publication.
--

VERSION 2 – AUTHOR RESPONSE

Reviewer: 1	
Thank you for the opportunity and accepting the suggestions given in the previous review. A few minor revisions like reframing the abstract introduction and other minor suggestions as commented will be appreciated. It is a great initiative and congratulations to the researchers as well as the BMJ for disseminating the research across the globe. ***Please also see attached comments***	Thank you. Attached comments have been addressed as follow;  • We reviewed the long sentence in page one and broke it to several sentences. • We re-write the number to words in page one. • We made grammatical corrections in page 4.

Reviewer: 2	
The paper is reading well and has shown great changes from the recommendations provided in the first review submission. I have two sections that i have noted which require alignment:	Thank you
 • 1. Section on Specific Objectives and Research Questions must be aligned and speak to each other. Specific objectives should translate into the specific research questions. 	Thank you for these observations, changes have been made to match objectives to research questions.
 • 2. Section on ETHICS AND DISSEMINATION - i have noted Tanzania appearing for the first time and please provide a justification on why the findings are important for Tanzania. 	Although we are planning to conduct follow-up research in Tanzania, we have decided to remove this information from the protocol as it is not necessary for the purpose of this review.”
If these issues are addressed, the manuscript is ready for publication.	Thank you